# HIGH VARIANCE SCORE FUNCTION ESTIMATES HELP DIFFUSION MODELS GENERALIZE

## ABSTRACT

How do diffusion-based generative models generalize beyond their training set? In particular, do they perform something similar to kernel density estimation? If so, what is the kernel, and which aspects of training and sampling determine its form? We argue that a key contributor to generalization is the fact that the denoising score matching objective usually used to train diffusion models tends to obtain high variance score function estimates at early times. We investigate this claim by mathematically studying score estimation for (unconditional) diffusion models using estimators that are linear in a set of feature maps. We show that, using standard choices (e.g., for the time sampling distribution), the effect of this high variance is mathematically equivalent to adding a noise term to the probability flow ODE. Moreover, in the special case that the score is learned independently for different times, reverse diffusion is on average equivalent to convolving the training distribution with a data-dependent kernel function.

## 1 INTRODUCTION

Despite their empirical successes, it is unclear how diffusion-based generative models (Sohl-Dickstein et al., 2015; Song & Ermon, 2019; Ho et al., 2020) are able to generalize. For example, how are image models able to strike a balance between generating images which are novel, and generating images which are like those from their training set?

Generating samples involves two steps: training a model, usually using a denoising score matching objective (Vincent, 2011; Song & Ermon, 2019); and sampling from that model, which can be viewed as numerically integrating an ordinary or stochastic differential equation (ODE/SDE) (Song et al., 2021). It is at least somewhat clear where generalization probably does *not* come from. Although noise in the sampling process can increase sample quality—as measured by, e.g., Fréchet inception distance (FID) scores—diffusion models can achieve high sample quality without this (Karras et al., 2022). Inaccuracies in numerical integration also do not appear to be responsible for generalization, since more precise integration (e.g., by using smaller time steps, or a more sophisticated numerical integration scheme) generally improves sample quality (Liu et al., 2022).

Generalization also does not appear to be due to perfectly optimizing the denoising score matching objective, since the optimal solution is the score function of the training distribution (Vincent, 2011). In particular, since models are trained using a finite number of examples, the optimal score function is that of a mixture of delta functions centered at the training data. Sampling using such a score function would only ever yield one of the training examples, rather than a novel sample (see Appendix A for a quick review of these points).

Finally, although function approximators like neural networks exhibit interesting inductive biases in what they readily learn from training data (Bordelon et al., 2020; Canatar et al., 2021), the architectures supporting large models like Stable Diffusion (Rombach et al., 2022) are flexible enough to in principle learn something extremely close to the optimal score. Although the inductive biases of neural networks are probably part of the story, it is unlikely that these types of inductive biases alone are responsible for generalization.

Where, then, might the ability to generalize come from? In this paper, we examine the possibility that generalization ability arises at least in part from using an objective function whose optimization typically produces high variance estimates of the score function. We mathematically show that

this high variance effectively contributes a (generally state- and time-dependent) noise term to the probability flow ODE. Moreover, the form of the kernel that appears in this noise term appears to have properties that support generalization, apparently by implementing an inductive bias about feature variance.

## 2 MATHEMATICAL FORMULATION

**Diffusion models.** Our mathematical formulation of diffusion models will be similar to that of Song et al. (2021). Training data from a distribution $p(\boldsymbol{x}_0)$ on $\mathbb{R}^D$ is corrupted by a forward process, producing a distribution of corrupted data $p(\boldsymbol{x}_t) := \int p(\boldsymbol{x}_t|\boldsymbol{x}_0)p(\boldsymbol{x}_0)\,d\boldsymbol{x}_0$. Data can be 'denoised' using a probability flow ODE involving the score function $\boldsymbol{s}(\boldsymbol{x}_t, t) := \nabla_{\boldsymbol{x}_t} \log p(\boldsymbol{x}_t)$; concretely,

$$\dot{\boldsymbol{x}}_t = -\beta_t \boldsymbol{x}_t + g_t \boldsymbol{\eta}_t \qquad \text{(forward process, integrate from } t=0 \text{ to } t=t_{max}) \tag{1}$$

$$\dot{\boldsymbol{x}}_t = -\beta_t \boldsymbol{x}_t - \frac{1}{2}g_t^2 \boldsymbol{s}(\boldsymbol{x}_t, t) \qquad \text{(reverse process, integrate from } t=t_{max} \text{ to } t=0) \tag{2}$$

where $\boldsymbol{\eta}_t \in \mathbb{R}^D$ is Gaussian white noise, and both $\beta_t > 0$ and $g_t > 0$ are smooth functions of $t \in [0, t_{max}]$. The forward process' marginals are $p(\boldsymbol{x}_t|\boldsymbol{x}_0) = \mathcal{N}(\boldsymbol{x}_t; \alpha_t \boldsymbol{x}_0, \sigma_t^2 \boldsymbol{I})$, where

$$\alpha_t := e^{-\int_0^t \beta_{t'}\,dt'} \qquad\qquad \sigma_t^2 := \int_0^t g_{t'}^2 \alpha_{t'}^2\,dt' . \tag{3}$$

One sometimes assumes specific relationships between the functions above; for example, the variance-preserving SDE (VP SDE) assumes $\beta_t = g_t^2/2$, so that $\alpha_t^2 + \sigma_t^2 = 1$ for all times $t$. In what follows, we will make no such assumptions.

**Denoising score matching.** The naive approach to learning a score estimator $\hat{\boldsymbol{s}}_{\boldsymbol{\theta}}(\boldsymbol{x}_t, t)$ might use

$$J_0(\boldsymbol{\theta}) := \frac{1}{2}\mathbb{E}_{t,\boldsymbol{x}_t|t}\left\{\|\hat{\boldsymbol{s}}_{\boldsymbol{\theta}}(\boldsymbol{x}_t, t) - \boldsymbol{s}(\boldsymbol{x}_t, t)\|_2^2\right\} = \frac{1}{2}\int \lambda(t)\|\hat{\boldsymbol{s}}_{\boldsymbol{\theta}}(\boldsymbol{x}_t, t) - \boldsymbol{s}(\boldsymbol{x}_t, t)\|_2^2\, p(\boldsymbol{x}_t)\,d\boldsymbol{x}_t dt \tag{4}$$

where $\lambda(t)$ is a time sampling distribution on $[0, t_{max}]$. In practice, only samples from $p(\boldsymbol{x}_0)$ are available, so it can be difficult to estimate $\boldsymbol{s}(\boldsymbol{x}_t, t)$; we can avoid this via the denoising score matching (DSM) objective (Vincent, 2011; Song & Ermon, 2019)

$$\begin{aligned} J_1(\boldsymbol{\theta}) &:= \frac{1}{2}\mathbb{E}_{t,\boldsymbol{x}_0,\boldsymbol{x}_t|\{\boldsymbol{x}_0,t\}}\left\{\|\hat{\boldsymbol{s}}_{\boldsymbol{\theta}}(\boldsymbol{x}_t, t) - \boldsymbol{s}(\boldsymbol{x}_t, t; \boldsymbol{x}_0)\|_2^2\right\} \\ &= \frac{1}{2}\int \lambda(t)\,\|\hat{\boldsymbol{s}}_{\boldsymbol{\theta}}(\boldsymbol{x}_t, t) - \nabla_{\boldsymbol{x}_t}\log p(\boldsymbol{x}_t|\boldsymbol{x}_0)\|_2^2\, p(\boldsymbol{x}_t|\boldsymbol{x}_0)p(\boldsymbol{x}_0)\,d\boldsymbol{x}_t d\boldsymbol{x}_0 dt . \end{aligned} \tag{5}$$

Both the naive and DSM objectives have the same optima (see Appendix A); however, before an optimal solution is reached, the variance of the score function estimates obtained using each objective are substantially different. Heuristically, this is because the proxy score function target has a large variance—a *singular* variance, in fact—for small times $\Delta t$:

$$\text{Cov}_{\boldsymbol{x}_t}\left[\nabla_{\boldsymbol{x}_t}\log p(\boldsymbol{x}|\boldsymbol{x}_0)\right] = \frac{1}{\sigma_t^2}\boldsymbol{I} \xrightarrow{t \to 0} \frac{1}{g_0^2 \Delta t}\boldsymbol{I} . \tag{6}$$

This means that the score function is typically being fit to close-to-random noise at small times $t$. Meanwhile, it is clear that it is relatively easy to estimate the score function at large times, since it is increasingly true that $p(\boldsymbol{x}_t) \approx p(\boldsymbol{x}_t|\boldsymbol{x}_0)$ as $t \to \infty$. We are not the first to identify this issue (Nguyen et al., 2017; Dhariwal & Nichol, 2021). For example, Chao et al. (2022) call this the *score mismatch* issue and propose a particular method for mitigating it. We take a different perspective here; we view this property not as a *bug*, but as a *feature* that may help diffusion models generalize.

**Time sampling and variance normalization.** In order to mitigate the large variance issue, practitioners usually do two things: (i) choose a special time sampling distribution $\lambda(t)$, and (ii) assume a score estimator with a certain $\sigma_t$-dependent prefactor. In particular, usually the distribution

$$\lambda_*(t) = \frac{\sigma_t^2}{\int_0^{t_{max}} \sigma_t^2\,dt} = \frac{\sigma_t^2}{Z_\sigma} \tag{7}$$

is used (Song et al., 2021; Karras et al., 2022), and we often take $\hat{\boldsymbol{s}}_{\boldsymbol{\theta}}(\boldsymbol{x}_t, t) = \boldsymbol{\epsilon}_{\boldsymbol{\theta}}(\boldsymbol{x}_t, t)/\sigma_t$. Then

$$J_1(\boldsymbol{\theta}) = \frac{1}{2Z_\sigma}\int \|\boldsymbol{\epsilon}_{\boldsymbol{\theta}}(\alpha_t \boldsymbol{x}_0 + \sigma_t \boldsymbol{\epsilon}, t) - \boldsymbol{\epsilon}\|_2^2\, \mathcal{N}(\boldsymbol{\epsilon}; \boldsymbol{0}, \boldsymbol{I})p(\boldsymbol{x}_0)\,d\boldsymbol{\epsilon} d\boldsymbol{x}_0 dt . \tag{8}$$

These choices are made by Stable Diffusion (Rombach et al., 2022), and similar choices are recommended by Karras et al. (2022).

## 3 GENERALIZATION AND SAMPLING VARIANCE: INTUITION

In supervised learning settings, "generalization" usually means predicting the value of a function on unseen inputs. It is critical to note that we mean something different when we refer to the ability of diffusion models to generalize. Real training data typically consists of $M \geq 1$ examples (e.g., images), which together define a mixture distribution:

$$p(\boldsymbol{x}_0) = \frac{1}{M} \sum_{m=1}^{M} \delta(\boldsymbol{x}_0 - \boldsymbol{\mu}_m) \qquad p(\boldsymbol{x}_t) = \frac{1}{M} \sum_{m=1}^{M} \mathcal{N}(\boldsymbol{x}_t; \alpha_t \boldsymbol{\mu}_m, \sigma_t^2 \boldsymbol{I}) \qquad (9)$$

where $\delta$ is the Dirac delta function. For such a distribution, we can straightforwardly compute that

$$\boldsymbol{s}(\boldsymbol{x}_t, t) = \frac{1}{M p(\boldsymbol{x}_t)} \sum_{m=1}^{M} \frac{(\alpha_t \boldsymbol{\mu}_m - \boldsymbol{x}_t)}{\sigma_t^2} \mathcal{N}(\boldsymbol{x}_t; \alpha_t \boldsymbol{\mu}_m, \sigma_t^2 \boldsymbol{I}) . \qquad (10)$$

What we mean by "generalization" is that our score estimator learns something *different* than this score function. Reverse diffusion with this score—the 'empirical' score—produces a sample from $p(\boldsymbol{x}_0)$, i.e., one of the $M$ training examples. What we would like instead is to generate samples similar to, but somewhat different from, those training examples. How might this be possible?

Given a large number of samples $(t^{(k)}, \boldsymbol{x}_0^{(k)}, \boldsymbol{x}^{(k)}) \sim \lambda(t) p(\boldsymbol{x}_0) p(\boldsymbol{x}_t | \boldsymbol{x}_0)$, we expect that a sufficiently expressive score estimator trained using a procedure like DSM is unbiased (since its optimum is the true score), i.e., that $\mathbb{E}[\hat{\boldsymbol{s}}_{\boldsymbol{\theta}}(\boldsymbol{x}_t, t)] = \boldsymbol{s}(\boldsymbol{x}_t, t)$, where the expectation is taken over sample realizations. The distribution learned by the diffusion model (obtained by reverse diffusion using the score estimator) can be written as $q(\boldsymbol{x}_0 | \boldsymbol{\theta}) = \int q(\boldsymbol{x}_0 | \boldsymbol{x}_T, \boldsymbol{\theta}) p(\boldsymbol{x}_T) \, d\boldsymbol{x}_T$, where $p(\boldsymbol{x}_T)$ is the distribution of the initial sample, and $q(\boldsymbol{x}_0 | \boldsymbol{x}_T, \boldsymbol{\theta})$ describes how that sample changes due to reverse diffusion. The function $q(\boldsymbol{x}_0 | \boldsymbol{x}_T, \boldsymbol{\theta})$ has a path integral representation (see Appendix B)

$$q(\boldsymbol{x}_0 | \boldsymbol{x}_T, \boldsymbol{\theta}) = \int \mathcal{D}[\boldsymbol{p}(t)] \mathcal{D}[\boldsymbol{x}(t)] \exp \left\{ \int_0^{t_{max}} i \boldsymbol{p}(t) \cdot \left[ \dot{\boldsymbol{x}}(t) + \beta_t \boldsymbol{x}_t + \frac{1}{2} g_t^2 \hat{\boldsymbol{s}}_{\boldsymbol{\theta}}(\boldsymbol{x}_t, t) \right] dt \right\}. \qquad (11)$$

If we take an expectation over sample realizations and parameter initializations, and hence compute the distribution 'typically' learned by an unbiased diffusion model, we obtain

$$\mathbb{E}[q(\boldsymbol{x}_0 | \boldsymbol{x}_T, \boldsymbol{\theta})] = \int \mathcal{D}[\boldsymbol{p}(t)] \mathcal{D}[\boldsymbol{x}(t)] \exp \left\{ M_1 + M_2 + \cdots \right\}$$

$$M_1 := \int_0^{t_{max}} i \boldsymbol{p}(t) \cdot \left[ \dot{\boldsymbol{x}}(t) + \beta_t \boldsymbol{x}_t + \frac{1}{2} g_t^2 \boldsymbol{s}(\boldsymbol{x}_t, t) \right] dt$$

$$M_2 := -\frac{1}{2} \int_0^{t_{max}} \int_0^{t_{max}} \left( \frac{g_t^2}{2} \cdot \frac{g_{t'}^2}{2} \right) \boldsymbol{p}(t)^T \mathrm{Cov}(\boldsymbol{s}_{\boldsymbol{\theta}}(\boldsymbol{x}_t, t), \boldsymbol{s}_{\boldsymbol{\theta}}(\boldsymbol{x}_{t'}, t')) \boldsymbol{p}(t') dt dt' .$$

The first term, $M_1$, is a 'mean' term which by itself corresponds to integrating the probability flow ODE. The second term, $M_2$, is a 'variance' term, which together with $M_1$ represents SDE dynamics with noise which is generically correlated across different states and times. One way for models to generalize is if $M_2$ is not negligible, since the added variance effectively produces a 'smeared out' version of the training distribution. Of course, we prefer *particular* smearings over others; for example, instead of smearing out each training data point independently of the others, we might prefer that the regions between data points receive additional probability.

But since constructing the score estimator involves averaging $N \gg 1$ samples, its covariance typically goes like $1/N$. In order for generalization to occur (in the absence of other effects, like early stopping) in the large $N$ limit, we need the covariance matrix to be $\mathcal{O}(1)$, and hence somewhat singular. Our claim is that this is happens for the DSM objective when certain choices are made, but *not* for the naive objective. Intuitively, this is due to the high variance of the score target used. In the next section, we will make this intuition mathematically precise.

## 4 MAIN THEORETICAL RESULTS

We are now ready to state our main theoretical results. For reasons of mathematical tractability, we consider score estimator that is linear in features (but not necessarily in $\boldsymbol{x}_t$ or $t$!)

$$\hat{\boldsymbol{s}}_{\boldsymbol{\theta}}(\boldsymbol{x}_t, t) = \frac{1}{\sigma_t}\left[\boldsymbol{w}_0 + \boldsymbol{W}\boldsymbol{\phi}(\boldsymbol{x}_t, t)\right] \qquad \hat{s}_i(\boldsymbol{x}_t, t) = \frac{1}{\sigma_t}\left[W_{i0} + \sum_{j=1}^{F} W_{ij}\phi_j(\boldsymbol{x}_t, t)\right], \quad (12)$$

where the feature maps $\boldsymbol{\phi} = (\phi_1, ..., \phi_F)^T$ are smooth functions from $\mathbb{R}^D \times [0, t_{max}]$ to $\mathbb{R}$ that are square-integrable with respect to $p(\boldsymbol{x}_t|\boldsymbol{x}_0)p(\boldsymbol{x}_0)\lambda(t)/\sigma_t^2$ and $p(\boldsymbol{x}_t|\boldsymbol{x}_0)p(\boldsymbol{x}_0)$ for all $t$. The parameters to be estimated are $\boldsymbol{\theta} := \{\boldsymbol{w}_0, \boldsymbol{W}\}$, with $\boldsymbol{w}_0 \in \mathbb{R}^D$ and $\boldsymbol{W} \in \mathbb{R}^{D \times F}$. We may abuse notation and write $\hat{\boldsymbol{s}}_{\boldsymbol{\theta}} = \boldsymbol{W}\boldsymbol{\phi}$, defining $W_{i0} := w_{0,i}$ and $\phi_0 := 1$ to absorb the constant term.

Denote the distribution of reverse diffusion outputs (see Eq. 11) by $q(\boldsymbol{x}_0|\boldsymbol{x}_T, \boldsymbol{\theta})$, the optimal parameters by $\boldsymbol{\theta}^* = \{\boldsymbol{w}_0^*, \boldsymbol{W}^*\}$, the optimal score estimator by $\boldsymbol{s}_*(\boldsymbol{x}_t, t)$, and the result of reverse diffusion using the optimal estimator by $q_*(\boldsymbol{x}_0)$. *What is the distribution $q$ 'typically' learned?*

In order to state our main result, it is useful to define the kernel matrices

$$\bar{K}_{ij} := \mathbb{E}_{t, \boldsymbol{x}_t|t}[\phi_i(\boldsymbol{x}_t, t)\phi_j(\boldsymbol{x}_t, t)/\sigma_t^2] \qquad \bar{K}_{ij}(0) := \mathbb{E}_{\boldsymbol{x}_0}[\phi_i(\boldsymbol{x}_0, 0)\phi_j(\boldsymbol{x}_0, 0)]. \quad (13)$$

**Theorem 1 (Linear score estimators trained via DSM asymptotically generalize)** *Suppose the parameters of a linear score estimator (Eq. 12) are optimized according to the DSM objective (Eq. 5) using $N$ independent samples from $\lambda_*(t)p(\boldsymbol{x}_0)p(\boldsymbol{x}_t|\boldsymbol{x}_0)$ (see Eq. 7). Consider the result of reverse diffusion using this estimator by Euler-integrating the probability flow ODE (Eq. 1) with a small time step $\Delta t$. If $N \to \infty$ and $\Delta t \to 0$ with $N\Delta t = c \gg 1$ held constant, then sampling from $\mathbb{E}[q(\boldsymbol{x}_0|\boldsymbol{x}_T, \boldsymbol{\theta})]$ is approximately equivalent to simulating the backwards-time (Ito-interpreted) SDE*

$$\dot{\boldsymbol{x}}_t = -\beta_t \boldsymbol{x}_t - \frac{1}{2}g_t^2 \boldsymbol{s}_*(\boldsymbol{x}_t, t) + \boldsymbol{\xi}(\boldsymbol{x}_t, t) \quad (14)$$

*from $t = t_{max}$ to $t = 0$ with initial condition $\boldsymbol{x}(t_{max}) = \boldsymbol{x}_T$. The noise term $\boldsymbol{\xi}(\boldsymbol{x}_t, t)$ has mean zero, and is generically correlated across different states and times according to*

$$Cov_{t,t',\boldsymbol{x}_t|t,\boldsymbol{x}_{t'}|t'}[\xi_i(\boldsymbol{x}_t, t), \xi_j(\boldsymbol{x}_{t'}, t')] = V_{ij}(\boldsymbol{x}_t, t, \boldsymbol{x}_{t'}, t') \quad (15)$$

*where we define the $D \times D$ "V kernel" $\boldsymbol{V}$ via*

$$V_{ij} := \frac{\delta_{ij}}{g_0^2 Z_\sigma c}\left(\frac{g_t^2}{2\sigma_t} \cdot \frac{g_{t'}^2}{2\sigma_{t'}}\right)\boldsymbol{\phi}(\boldsymbol{x}_t, t)^T \bar{\boldsymbol{K}}^{-1}\bar{\boldsymbol{K}}(0)\bar{\boldsymbol{K}}^{-1}\boldsymbol{\phi}(\boldsymbol{x}_{t'}, t'). \quad (16)$$

See Appendix C for the proof. One important corollary follows from the details of the argument:

**Corollary 1.1 (Linear score estimators trained via naive objective do not generalize)**
*Consider the situation described in Theorem 1, but assume parameters are instead optimized according to the naive objective (Eq. 4). We instead have $\mathbb{E}[q(\boldsymbol{x}_0|\boldsymbol{x}_T, \boldsymbol{\theta})] = q_*(\boldsymbol{x}_0|\boldsymbol{x}_T)$.*

The argument also provides insight about the requirements for generalization given DSM training:

**Corollary 1.2 (Generalization requires time sampling distribution to undersample small times)**
*Consider the situation described in Theorem 1, but assume the sampling distribution $\lambda(t)$ used has*

$$\lim_{\Delta t \to 0} \frac{\lambda(\Delta t)}{\Delta t} = 0. \quad (17)$$

*We instead have $\mathbb{E}[q(\boldsymbol{x}_0|\boldsymbol{x}_T, \boldsymbol{\theta})] = q_*(\boldsymbol{x}_0|\boldsymbol{x}_T)$.*

### 4.1 SPECIAL CASES

Some special cases can be worked out. Of particular interest is the case of gradient-descent-trained neural networks in the neural tangent kernel (NTK) regime (Jacot et al., 2018; Bietti & Mairal, 2019), where learning is 'lazy' (Chizat et al., 2019) in the sense that weights do not move much from their initial values.

Assume such a neural network parameterizes $\epsilon_{\boldsymbol{\theta}}(\boldsymbol{x}_t, t) = \sigma_t \hat{\boldsymbol{s}}_{\boldsymbol{\theta}}(\boldsymbol{x}_t, t)$. For such networks, since

$$\epsilon(\boldsymbol{x}_t, t; \boldsymbol{\theta}) \approx \epsilon(\boldsymbol{x}_t, t; \boldsymbol{\theta}_0) + \frac{\partial \epsilon(\boldsymbol{x}_t, t; \boldsymbol{\theta}_0)}{\partial \boldsymbol{\theta}_0}(\boldsymbol{\theta} - \boldsymbol{\theta}_0) \ , \tag{18}$$

i.e., the learned parameters $\boldsymbol{\theta}$ are not far from the initial parameters $\boldsymbol{\theta}_0$, we are in the linear regime described by Eq. 12 and Theorem 1 holds.

**Corollary 1.3 (NTK regime neural networks trained via DSM asymptotically generalize)**
*Consider the situation described in Theorem 1, except that $\epsilon_{\boldsymbol{\theta}}(\boldsymbol{x}_t, t)$ is a gradient-descent-trained neural network in the NTK regime. Then the conclusion of Theorem 1 still holds. (We assume normally distributed initial weights with the typical layer width scaling. The infinite width limit must be taken before the $N \to \infty$ limit.)*

The feature maps are usually difficult to write down explicitly (and in this context, it is more convenient to work with them than the NTK), but there are methods to construct them (see, e.g., Bietti & Mairal (2019)).

Another case of interest is what we discussed in the previous section: when the training distribution consists of $M$ examples, so that $p(\boldsymbol{x}_0)$ is a mixture of delta functions. Near $t = 0$, the score function produces an infinitely strong 'attractive force' towards one of the examples, so one may not expect the additional variance we have discussed to be much help. But it turns out that reproducing training data is avoided since the V kernel is *also* singular near $t = 0$; this effectively adds a 'convolution' step to the end of reverse diffusion.

**Corollary 1.4 (Mixture training set produces original distribution convolved with Gaussian kernel)**
*Consider the situation described in Theorem 1, except that $p(\boldsymbol{x}_0)$ is a mixture of $M$ delta functions centered on training data $\{\boldsymbol{\mu}_m\}_{m=1,\dots,M}$. We have that*

$$\mathbb{E}[q(\boldsymbol{x}_0|\boldsymbol{\theta})] = \sum_{m=1}^{M} w_m \mathcal{N}(\boldsymbol{x}_0; \boldsymbol{\mu}_m, \boldsymbol{V}(\boldsymbol{\mu}_m)) \tag{19}$$

*where $w_1 + \dots + w_M = 1$, but the weights are not necessarily the same as those of the original training distribution. In this case, the V kernel has a special form:*

$$\boldsymbol{V}(\boldsymbol{y}) := \frac{1}{4cZ_\sigma} \left[ 1 + (\boldsymbol{\phi}(\boldsymbol{y}, 0) - \boldsymbol{\mu}_{\boldsymbol{\phi}})^T \boldsymbol{\Sigma}_{\boldsymbol{\phi}}^{-1} (\boldsymbol{\phi}(\boldsymbol{y}, 0) - \boldsymbol{\mu}_{\boldsymbol{\phi}}) \right] \boldsymbol{I}$$

$$\boldsymbol{\mu}_{\boldsymbol{\phi}} := \mathbb{E}_{\boldsymbol{x}_0} \left[ \boldsymbol{\phi}(\boldsymbol{x}_0, 0) \right] \qquad \boldsymbol{\Sigma}_{\boldsymbol{\phi}} := \mathbb{E}_{\boldsymbol{x}_0} \left[ \boldsymbol{\phi}(\boldsymbol{x}_0, 0) \boldsymbol{\phi}(\boldsymbol{x}_0, 0)^T \right] - \boldsymbol{\mu}_{\boldsymbol{\phi}} \boldsymbol{\mu}_{\boldsymbol{\phi}}^T \tag{20}$$

Finally, a somewhat artificial but simple special case assumes that the parameters of the score function for different times are learned independently, e.g., via a sample-splitting scheme. In this case, $\mathbb{E}[q(\boldsymbol{x}_0|\boldsymbol{\theta})]$ is equivalent to running reverse diffusion with the optimal score, and then convolving the result with the ($t = 0$) V kernel. If the empirical score is learned perfectly, this has the effect of 'smearing out' the original data distribution.

## 5 INTERPRETING THE V KERNEL

A priori, it is unclear how to think about the potential utility—if any—of the V kernel. In this section, we will attempt to build intuition about it by examining several of its properties. We will focus on the special case that score function parameters for different times are learned independently.

### 5.1 EFFECT ON KERNEL ON MEAN AND VARIANCE

Let $\boldsymbol{\mu}_*$ and $\boldsymbol{\Sigma}_*$ denote the mean and covariance of the optimal distribution $q_*$. A typical learned distribution $q$ has the same mean as $q_*$, since

$$\int \boldsymbol{x}_0\, \mathcal{N}(\boldsymbol{x}_0; \boldsymbol{y}, \boldsymbol{V}(\boldsymbol{y}))\, q_*(\boldsymbol{y})\, d\boldsymbol{x}_0 d\boldsymbol{y} = \int \boldsymbol{y}\, q_*(\boldsymbol{y})\, d\boldsymbol{y} = \boldsymbol{\mu}_* \,. \tag{21}$$

On the other hand, it will have more variance than $q_*$, since

$$\int \boldsymbol{x}_0 \boldsymbol{x}_0^T\, \mathcal{N}(\boldsymbol{x}_0; \boldsymbol{y}, \boldsymbol{V}(\boldsymbol{y}))\, q_*(\boldsymbol{y})\, d\boldsymbol{x}_0 d\boldsymbol{y}$$

$$= \int \left\{ \boldsymbol{y}\boldsymbol{y}^T + \frac{Z_\sigma}{4c} \left[ 1 + (\boldsymbol{\phi}(\boldsymbol{y}) - \boldsymbol{\mu}_\phi)^T \boldsymbol{\Sigma}_\phi^{-1} (\boldsymbol{\phi}(\boldsymbol{y}) - \boldsymbol{\mu}_\phi) \right] \boldsymbol{I} \right\} q_*(\boldsymbol{y})\, d\boldsymbol{y} \tag{22}$$

$$= \boldsymbol{\Sigma}_* + \boldsymbol{\mu}_* \boldsymbol{\mu}_*^T + \frac{Z_\sigma}{4c}(1 + D)\boldsymbol{I} \,.$$

### 5.2 EXAMPLE: LINEAR FEATURES

Suppose that the score is estimated using linear features, i.e., $\boldsymbol{\phi} = (1, -x_1, ..., -x_D)^T$ so that $F = D$. The optimal distribution is $q_* = \mathcal{N}(\boldsymbol{\mu}, \boldsymbol{\Sigma})$, where $\boldsymbol{\mu}$ and $\boldsymbol{\Sigma}$ denote the sample mean and covariance. The V kernel is

$$\boldsymbol{V}(\boldsymbol{y}) = \frac{Z_\sigma}{4c} \left[ 1 + (\boldsymbol{y} - \boldsymbol{\mu})^T \boldsymbol{\Sigma}^{-1} (\boldsymbol{y} - \boldsymbol{\mu}) \right] \boldsymbol{I} \,. \tag{23}$$

Note that the $\boldsymbol{y}$-dependent term is small if $\boldsymbol{y}$ is close to the mean, and large if it is at least one standard deviation away from it along some state space direction.

It is worth noting that convolving $q_*$ with the V kernel does not generally produce another distribution from the same family; in this case, a typical $q$ is not Gaussian. One way to see this is via its characteristic function:

$$\psi(\boldsymbol{u}) = \int e^{i\boldsymbol{u}\cdot\boldsymbol{x}_0}\, \mathcal{N}(\boldsymbol{x}_0; \boldsymbol{y}, \boldsymbol{V}(\boldsymbol{y}))\, q_*(\boldsymbol{y})\, d\boldsymbol{x}_0 d\boldsymbol{y}$$

$$= \int e^{i\boldsymbol{u}\cdot\boldsymbol{y} - \frac{\boldsymbol{u}\cdot\boldsymbol{u}}{2} \frac{Z_\sigma}{4c}\left[1 + (\boldsymbol{y}-\boldsymbol{\mu})^T\boldsymbol{\Sigma}^{-1}(\boldsymbol{y}-\boldsymbol{\mu})\right]}\, q_*(\boldsymbol{y})\, d\boldsymbol{y} \tag{24}$$

$$= \frac{1}{\left[1 + \frac{Z_\sigma}{4c}\boldsymbol{u}\cdot\boldsymbol{u}\right]^{D/2}} \exp\left\{ i\boldsymbol{u}\cdot\boldsymbol{\mu} - \frac{\boldsymbol{u}^T\boldsymbol{\Sigma}\boldsymbol{u}}{2} \frac{1}{1 + \frac{Z_\sigma}{4c}\boldsymbol{u}\cdot\boldsymbol{u}} \right\} \,.$$

In particular, the log-characteristic function is not quadratic, but involves higher order terms that depend on powers of $1/c$.

### 5.3 EXAMPLE: ORTHOGONAL FEATURES

Assume that the estimator uses features which are orthogonal with respect to the data, in the sense that the covariance matrix $\boldsymbol{\Sigma}_\phi$ is diagonal. Then the V kernel involves a sum of squared feature

norms weighted by feature variances. This suggests the same intuition as for the previous normal distribution example: a large amount of noise is added to regions of state space where features are far from their 'typical' values, and a small amount of noise is added to regions of state space where features are typical.

An interesting special case is when features correspond to non-overlapping bins with the same amplitude (1, say). In that case, the diagonal entries of the feature covariance matrix are proportional to

$$\frac{1 - p_i}{p_i} \ , \tag{25}$$

where $p_i$ is the probability in the training data captured by the $i$-th bin. More noise is added where bins capture less of the overall probability. In the case where each data point is associated with exactly one bin, the same amount of noise is added to each of those points.

## 5.4 EXAMPLE: GAUSSIAN MIXTURE FEATURES

A Gaussian mixture with $M$ mixture components has

$$p(\boldsymbol{x}_0) = \sum_{m=1}^{M} w_m \mathcal{N}(\boldsymbol{x}_0; \boldsymbol{\mu}_m, \boldsymbol{\Sigma}_m) \qquad \boldsymbol{s}(\boldsymbol{x}_0) = \frac{w_m \boldsymbol{\Sigma}_m^{-1}(\boldsymbol{\mu}_m - \boldsymbol{x}_0)\mathcal{N}(\boldsymbol{x}_0; \boldsymbol{\mu}_m, \boldsymbol{\Sigma}_m)}{\sum_{r=1}^{M} w_r \mathcal{N}(\boldsymbol{x}_0; \boldsymbol{\mu}_r, \boldsymbol{\Sigma}_r)} \ . \tag{26}$$

Attempting to estimate all parameters of a Gaussian mixture yields a problem not linear in the sense we consider in this paper; however, one can study an analogous linear problem by defining features

$$\boldsymbol{\phi}_m(\boldsymbol{x}) = \frac{(\boldsymbol{\mu}_m - \boldsymbol{x})}{\sigma^2} \frac{\mathcal{N}(\boldsymbol{x}; \boldsymbol{\mu}_m, \sigma^2 \boldsymbol{I})}{\sum_r \mathcal{N}(\boldsymbol{x}; \boldsymbol{\mu}_r, \sigma^2 \boldsymbol{I})} \tag{27}$$

and fitting the score estimator

$$\hat{\boldsymbol{s}}_{\boldsymbol{\theta}}(\boldsymbol{x}_t, t) = \boldsymbol{W}_0(t) + \sum_{m=1}^{M} \boldsymbol{W}_m(t) \boldsymbol{\phi}_m(\boldsymbol{x}_t) \tag{28}$$

to a ground truth distribution $p(\boldsymbol{x}_0) = \frac{1}{M} \sum_m \mathcal{N}(\boldsymbol{x}_0; \boldsymbol{\mu}_m, \sigma^2 \boldsymbol{I})$. In this case, the matrix $\boldsymbol{\Sigma}_{\boldsymbol{\phi}}$ whose inverse appears in the V kernel has a special meaning; since $\boldsymbol{\phi}_m(\boldsymbol{x}_0) = -\nabla_{\boldsymbol{\mu}_m} \log p(\boldsymbol{x}_0)$, the $\boldsymbol{\phi}$ covariance matrix is precisely the Fisher information matrix associated with the ground truth distribution. Given that the Fisher information matrix fundamentally bounds how well score function parameters can be estimated, the V kernel convolution appears to apply additional 'smearing' to observations in regions of state space where the score function is insensitive to small changes in its parameters, and less 'smearing' in regions where it is highly sensitive to small parameter changes.

## 5.5 GENERAL ROLE OF THE V KERNEL

In general, it may be useful to think of the V kernel as something which implements particular inductive biases which may be useful for generalization. One is the bias that features tend to take typical values, and that data for which this is not true should be considered less reliable. Another is that the structure of the feature space used by the estimator somehow reflects the true data distribution; for example, assuming non-overlapping bins yields a different kind of kernel (where different points are treated identically) than assuming overlapping ones (where points can affect one another). Allowing data points to interact by having at least one feature take a non-negligible value on both may be important for, e.g., interpolation.

## 6 DISCUSSION AND CONCLUSION

We were able to show mathematically that, contrary to what one might expect, there is a sense in which diffusion models tend to learn something other than the optimum of the objective they are trained on; moreover, we were able to show that the learned distribution is mathematically equivalent to convolving the optimal distribution with a particular kernel function.

The defining property of the kernel function is that it adds a large amount of noise to data in regions where at least one feature direction is somewhat larger than expected, and a small amount of noise in regions where features take more typical values (relative to their variances). From the perspective of constructing something like a kernel density estimate that generalizes well, this makes some intuitive sense. We ought to convolve data points with a kernel that corresponds to what we would *expect* to sample from that region of state space, if we were to sample from it again. Regions far from the mean are low in probability, and observations there should be 'smeared out' substantially; meanwhile, regions close to the mean are more reliable, and should not be.

There are a number of limitations of the current work related to scope. (i) Our calculation assumes an estimator that is linear in its features; (ii) we only consider *unconditional* models; (iii) no attention mechanism is included; (iv) and we do not consider learning dynamics. Nonetheless, it is our hope that this calculation provides a foundation for others to rigorously understand the inductive biases of diffusion models.

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
