## A    DENOISING SCORE MATCHING PRESERVES OPTIMA OF NAIVE OBJECTIVE

For ease of reference, we will show in this appendix that the naive objective

$$
\begin{aligned}
J_0(\boldsymbol{\theta}) &:= \frac{1}{2}\mathbb{E}_{t,\boldsymbol{x}_t|t}\left\{\|\hat{\boldsymbol{s}}_{\boldsymbol{\theta}}(\boldsymbol{x}_t,t) - \boldsymbol{s}(\boldsymbol{x}_t,t)\|_2^2\right\} \\
&= \frac{1}{2}\int \lambda(t)\|\hat{\boldsymbol{s}}_{\boldsymbol{\theta}}(\boldsymbol{x}_t,t) - \boldsymbol{s}(\boldsymbol{x}_t,t)\|_2^2\, p(\boldsymbol{x}_t)\, d\boldsymbol{x}_t dt
\end{aligned}
\tag{29}
$$

and the DSM objective

$$
\begin{aligned}
J_1(\boldsymbol{\theta}) &:= \frac{1}{2}\mathbb{E}_{t,\boldsymbol{x}_0,\boldsymbol{x}_t|\{t,\boldsymbol{x}_0\}}\left\{\|\hat{\boldsymbol{s}}_{\boldsymbol{\theta}}(\boldsymbol{x}_t,t) - \boldsymbol{s}(\boldsymbol{x}_t,t;\boldsymbol{x}_0)\|_2^2\right\} \\
&= \frac{1}{2}\int \lambda(t)\|\hat{\boldsymbol{s}}_{\boldsymbol{\theta}}(\boldsymbol{x}_t,t) - \nabla_{\boldsymbol{x}_t}\log p(\boldsymbol{x}_t|\boldsymbol{x}_0)\|_2^2\, p(\boldsymbol{x}_t|\boldsymbol{x}_0)p(\boldsymbol{x}_0)\, d\boldsymbol{x}_t d\boldsymbol{x}_0 dt
\end{aligned}
\tag{30}
$$

have the same optima. See Vincent (2011), Song & Ermon (2019), and Song et al. (2021) for more discussion of this point.

Assume that $\boldsymbol{x}_t, \boldsymbol{x}_0 \in \mathbb{R}^D$ and that $\boldsymbol{\theta} \in \mathbb{R}^P$. The gradient of $J_0$ with respect to $\boldsymbol{\theta}$ is

$$
\nabla_{\boldsymbol{\theta}} J_0 = \int \lambda(t)\left(\nabla_{\boldsymbol{\theta}}\hat{\boldsymbol{s}}_{\boldsymbol{\theta}}(\boldsymbol{x}_t,t)\right)^T \left[\hat{\boldsymbol{s}}_{\boldsymbol{\theta}}(\boldsymbol{x}_t,t) - \boldsymbol{s}(\boldsymbol{x}_t,t)\right]\, p(\boldsymbol{x}_t)\, d\boldsymbol{x}_t dt
\tag{31}
$$

where $\nabla_{\boldsymbol{\theta}}\hat{\boldsymbol{s}}_{\boldsymbol{\theta}}(\boldsymbol{x}_t,t)$ is the $D \times P$ Jacobian matrix of the score estimator. The gradient of $J_1$ is

$$
\nabla_{\boldsymbol{\theta}} J_1 = \int \lambda(t)\left(\nabla_{\boldsymbol{\theta}}\hat{\boldsymbol{s}}_{\boldsymbol{\theta}}(\boldsymbol{x}_t,t)\right)^T \left[\hat{\boldsymbol{s}}_{\boldsymbol{\theta}}(\boldsymbol{x}_t,t) - \boldsymbol{s}(\boldsymbol{x}_t,t;\boldsymbol{x}_0)\right]\, p(\boldsymbol{x}_t|\boldsymbol{x}_0)p(\boldsymbol{x}_0)\, d\boldsymbol{x}_t d\boldsymbol{x}_0 dt\ .
\tag{32}
$$

At this point, we will make two observations about the gradient of $J_1$. First, the term on the left does not depend on $\boldsymbol{x}_0$, so we can marginalize over $\boldsymbol{x}_0$. Explicitly,

$$
\begin{aligned}
&\int \lambda(t)\left(\nabla_{\boldsymbol{\theta}}\hat{\boldsymbol{s}}_{\boldsymbol{\theta}}(\boldsymbol{x}_t,t)\right)^T \hat{\boldsymbol{s}}_{\boldsymbol{\theta}}(\boldsymbol{x}_t,t)\, p(\boldsymbol{x}_t|\boldsymbol{x}_0)p(\boldsymbol{x}_0)\, d\boldsymbol{x}_t d\boldsymbol{x}_0 dt \\
&= \int \lambda(t)\left(\nabla_{\boldsymbol{\theta}}\hat{\boldsymbol{s}}_{\boldsymbol{\theta}}(\boldsymbol{x}_t,t)\right)^T \hat{\boldsymbol{s}}_{\boldsymbol{\theta}}(\boldsymbol{x}_t,t)\, p(\boldsymbol{x}_t)\, d\boldsymbol{x}_t dt\ .
\end{aligned}
\tag{33}
$$

Second, the term on the right only depends on $\boldsymbol{x}_0$ through the proxy score target. Moreover,

$$
\begin{aligned}
\int \boldsymbol{s}(\boldsymbol{x}_t,t;\boldsymbol{x}_0)p(\boldsymbol{x}_t|\boldsymbol{x}_0)p(\boldsymbol{x}_0)\, d\boldsymbol{x}_0 &= \int \nabla_{\boldsymbol{x}_t}\log p(\boldsymbol{x}_t|\boldsymbol{x}_0)p(\boldsymbol{x}_t|\boldsymbol{x}_0)p(\boldsymbol{x}_0)\, d\boldsymbol{x}_0 \\
&= \int \nabla_{\boldsymbol{x}_t}p(\boldsymbol{x}_t|\boldsymbol{x}_0)p(\boldsymbol{x}_0)\, d\boldsymbol{x}_0 \\
&= \nabla_{\boldsymbol{x}_t}\int p(\boldsymbol{x}_t|\boldsymbol{x}_0)p(\boldsymbol{x}_0)\, d\boldsymbol{x}_0 \\
&= \nabla_{\boldsymbol{x}_t}p(\boldsymbol{x}_t) \\
&= \boldsymbol{s}(\boldsymbol{x}_t,t)p(\boldsymbol{x}_t)\ .
\end{aligned}
\tag{34}
$$

Hence, the gradient of $J_0$ is the same as the gradient of $J_1$, so they have the same optima. If the score approximator is arbitrarily expressive, we in particular have that the true score (a global minimum of $J_0$) is an optimum of the DSM objective.

## B PATH INTEGRAL REPRESENTATION OF REVERSE DIFFUSION

A general (backwards-time) ODE—of which the probability flow ODE (Eq. 1) is just one example—can be written

$$\dot{\boldsymbol{x}}_t = \boldsymbol{f}(\boldsymbol{x}_t, t) . \tag{35}$$

Given an initial condition $\boldsymbol{x}_{t_{max}} = \boldsymbol{x}_T$ and Euler time steps of size $\Delta t$, the distribution of outputs $\boldsymbol{x}_0$ after numerically integrating for an amount of time $t_{max}$ is equal to

$$q(\boldsymbol{x}_0|\boldsymbol{x}_T) = \int d\boldsymbol{x}_1 \cdots d\boldsymbol{x}_{T-1}\, \delta(\boldsymbol{x}_0 = \boldsymbol{x}_1 - \boldsymbol{f}(\boldsymbol{x}_1, t_1)\Delta t) \cdots \delta(\boldsymbol{x}_{T-1} = \boldsymbol{x}_T - \boldsymbol{f}(\boldsymbol{x}_T, t_T)\Delta t)$$

where $t_\tau := \tau\Delta t$ and $t_T := t_{max}$. Using the standard Fourier representation of the delta function, we can write this as

$$q(\boldsymbol{x}_0|\boldsymbol{x}_T) = \int d\boldsymbol{x}_1 \cdots d\boldsymbol{x}_{T-1} \frac{d\boldsymbol{p}_0 \cdots d\boldsymbol{p}_{T-1}}{(2\pi)^T} \exp\left\{ \sum_{\tau=0}^{T-1} -i\boldsymbol{p}_\tau \cdot [\boldsymbol{x}_\tau - \boldsymbol{x}_{\tau+1} + \boldsymbol{f}(\boldsymbol{x}_{\tau+1}, t_{\tau+1})\Delta t] \right\} .$$

The integration measure is often written in shorthand form as

$$\mathcal{D}[\boldsymbol{p}(t)]\mathcal{D}[\boldsymbol{x}(t)] := d\boldsymbol{x}_1 \cdots d\boldsymbol{x}_{T-1} \frac{d\boldsymbol{p}_0 \cdots d\boldsymbol{p}_{T-1}}{(2\pi)^T} , \tag{36}$$

especially when we take $T$ large, but one should keep in mind that the path integral only remains well-behaved for finite $\Delta t$. (In other words, we prefer to transition to a different representation before we take $\Delta t$ completely to zero, for example.) When $\Delta t$ is small, we can express the argument of the exponential in the useful heuristic form

$$\int_0^{t_{max}} i\boldsymbol{p}(t) \cdot [\dot{\boldsymbol{x}}(t) - \boldsymbol{f}(\boldsymbol{x}(t), t)]\, dt . \tag{37}$$

We can obtain Eq. 11 by specializing this result to the particular choice of $\boldsymbol{f}$ that appears in the probability flow ODE.

This representation is particularly convenient for averaging over the weights, since averaging over them simply corresponds to evaluating their characteristic functions.

## C  OUTLINE OF PROOF TO MAIN ARGUMENT

We would like to prove Theorem 1 (see Sec. 4). We remind the reader that we are considering a linear score estimator

$$\hat{\boldsymbol{s}}_{\boldsymbol{\theta}}(\boldsymbol{x}_t, t) = \frac{1}{\sigma_t} \left[ \boldsymbol{w}_0 + \boldsymbol{W} \boldsymbol{\phi}(\boldsymbol{x}_t, t) \right] \qquad \hat{s}_i(\boldsymbol{x}_t, t) = \frac{1}{\sigma_t} \left[ W_{i0} + \sum_{j=1}^{F} W_{ij} \phi_j(\boldsymbol{x}_t, t) \right], \qquad (38)$$

where the feature maps $\boldsymbol{\phi} = (\phi_1, ..., \phi_F)^T$ are smooth functions from $\mathbb{R}^D \times [0, t_{max}]$ to $\mathbb{R}$ that are square-integrable with respect to $p(\boldsymbol{x}_t|\boldsymbol{x}_0, t)p(\boldsymbol{x}_0)\lambda(t)/\sigma_t^2$ and $p(\boldsymbol{x}_t|\boldsymbol{x}_0, t)p(\boldsymbol{x}_0)$ for all $t$. The parameters to be estimated are $\boldsymbol{\theta} := \{\boldsymbol{w}_0, \boldsymbol{W}\}$, with $\boldsymbol{w}_0 \in \mathbb{R}^D$ and $\boldsymbol{W} \in \mathbb{R}^{D \times F}$. We may abuse notation and write $\hat{\boldsymbol{s}}_{\boldsymbol{\theta}} = \boldsymbol{W} \boldsymbol{\phi}$, defining $W_{i0} := w_{0,i}$ and $\phi_0 := 1$ to absorb the constant term.

The overall idea is as follows. We would like to show that the variance of the optimal score estimator is singular, and that the singularity scales like $1/(\Delta t)$. Hence, if we assume $N\Delta t = \text{const.}$, there is a nonvanishing variance term in the path integral representation of reverse diffusion dynamics (see Eq. 11 and Appendix B). In order to do this, we must first compute the optimal score estimator (Appendix D), and compute its covariance (Appendix E).

### C.1  DSM OPTIMUM

It will be useful to define several matrices. Define the matrix $\boldsymbol{J} \in \mathbb{R}^{(F+1) \times D}$, the matrix $\boldsymbol{K} \in \mathbb{R}^{(F+1) \times (F+1)}$, and their averages:

$$J_{ij}(\boldsymbol{x}_t, \boldsymbol{x}_0, t) := -\frac{\phi_i(\boldsymbol{x}_t, t)}{\sigma_t} s_j(\boldsymbol{x}_t, t; \boldsymbol{x}_0) \qquad \bar{\boldsymbol{J}} := \mathbb{E}_{t, \boldsymbol{x}_0, \boldsymbol{x}_t | \{t, \boldsymbol{x}_0\}} \left[ \boldsymbol{J}(\boldsymbol{x}_t, \boldsymbol{x}_0, t) \right]$$

$$K_{ij}(\boldsymbol{x}_t, t) := \frac{\phi_i(\boldsymbol{x}_t, t)\phi_j(\boldsymbol{x}_t, t)}{\sigma_t^2} \qquad \bar{\boldsymbol{K}} := \mathbb{E}_{t, \boldsymbol{x}_t | t} \left[ \boldsymbol{K}(\boldsymbol{x}_t, t) \right] \qquad (39)$$

$$\bar{K}_{ij}(0) := \mathbb{E}_{\boldsymbol{x}_0}[\phi_i(\boldsymbol{x}_0, 0)\phi_j(\boldsymbol{x}_0, 0)] .$$

In defining the above matrices, the constant 1 is treated as the zeroth feature. It is worth noting that $\bar{\boldsymbol{K}}$ is a kind of kernel matrix; for example,

$$\bar{K}_{ij}(0) = \frac{1}{M} \sum_m \phi_i(\boldsymbol{\mu}_m, 0)\phi_j(\boldsymbol{\mu}_m, 0) \qquad (40)$$

given a training distribution $p(\boldsymbol{x}_0)$ which is a mixture of $M$ delta functions at locations $\boldsymbol{\mu}_m$. $\bar{\boldsymbol{J}}$ is a Jacobian, since one can show

$$\bar{J}_{ij} = \mathbb{E}_{t, \boldsymbol{x}_t | t} \left( \frac{1}{\sigma_t} \frac{\partial \phi_i(\boldsymbol{x}_t, t)}{\partial x_j} \right) \qquad (41)$$

via integration by parts. In Appendix D, we prove the following:

**Proposition 2 (DSM optimum in linear regression setting)** *Given a linear score estimator (Eq. 38), the optimum of the DSM objective (Eq. 5) is*

$$\boldsymbol{W}^* = -\bar{\boldsymbol{J}}^T \bar{\boldsymbol{K}}^{-1} \qquad (42)$$

*where $\boldsymbol{W}^*$ here includes the optimal weights of the constant feature. Separating the constant feature from the non-constant features, we have*

$$\boldsymbol{w}_0^* = \tilde{\boldsymbol{J}}^T \boldsymbol{\Sigma}_{\boldsymbol{\phi}}^{-1} \boldsymbol{\mu}_{\boldsymbol{\phi}} \qquad \boldsymbol{W}^* = -\tilde{\boldsymbol{J}}^T \boldsymbol{\Sigma}_{\boldsymbol{\phi}}^{-1} \qquad (43)$$

*where*

$$\boldsymbol{\mu}_{\boldsymbol{\phi}} := \frac{1}{t_{max}} \int_0^{t_{max}} \mathbb{E}_{\boldsymbol{x}_t | t} \left[ \boldsymbol{\phi}(\boldsymbol{x}_t, t) \right] dt$$

$$\boldsymbol{\Sigma}_{\boldsymbol{\phi}} := \frac{1}{t_{max}} \int_0^{t_{max}} \mathbb{E}_{\boldsymbol{x}_t | t} \left[ \boldsymbol{\phi}(\boldsymbol{x}_t, t)\boldsymbol{\phi}(\boldsymbol{x}_t, t)^T \right] dt - \boldsymbol{\mu}_{\boldsymbol{\phi}} \boldsymbol{\mu}_{\boldsymbol{\phi}}^T , \qquad (44)$$

*and where $\tilde{\boldsymbol{J}} \in \mathbb{R}^{F \times D}$ refers to $\bar{\boldsymbol{J}}$ with the zeroth row removed, and with an extra prefactor of $Z_\sigma/t_{max}$. The optimal score is*

$$\hat{\boldsymbol{s}}_*(\boldsymbol{x}_t, t) = \tilde{\boldsymbol{J}}^T \boldsymbol{\Sigma}_{\boldsymbol{\phi}}^{-1} \left[ \boldsymbol{\mu}_{\boldsymbol{\phi}} - \boldsymbol{\phi}(\boldsymbol{x}_t, t) \right] . \qquad (45)$$

### C.2 ASYMPTOTIC VARIANCE OF SCORE ESTIMATOR

Given the DSM optimum, 'training' a model using $N$ samples is equivalent to constructing sample mean estimators of $\bar{\boldsymbol{J}}$ and $\bar{\boldsymbol{K}}$.

The covariance of the sample mean estimators $\hat{\boldsymbol{J}}$ and $\hat{\boldsymbol{K}}$ determines the covariance of the score estimator. These covariances are derived in Appendix E and summarized in the proposition below:

**Proposition 3 (Asymptotic variance of score estimator)** *The following covariances are finite:*

$$Cov_{t,\boldsymbol{x}_t|t}(K_{ij}, K_{k\ell}) < \infty \qquad Cov_{t,\boldsymbol{x}_0,\boldsymbol{x}_t|\{t,\boldsymbol{x}_0\}}(J_{ij}, K_{k\ell}) < \infty \ . \tag{46}$$

*On the other hand, the following covariance is formally infinite (and in particular scales like $1/(\Delta t)$ when the relevant time integral is discretized):*

$$Cov_{t,\boldsymbol{x}_0,\boldsymbol{x}_t|\{t,\boldsymbol{x}_0\}}(J_{ij}, J_{k\ell}) = \frac{1}{Z_\sigma} \frac{\delta_{j\ell}}{g_0^2 \Delta t} \bar{K}_{ik}(0) \xrightarrow{\Delta t \to 0} \delta_{j\ell} \bar{K}_{ik}(0)\infty \ . \tag{47}$$

*But this is only true when the time sampling distribution has $\lambda(\Delta t) \propto_{\Delta t} \Delta t$ for sufficiently small $\Delta t$. (This is true of $\lambda_*(t) \propto_t \sigma_t^2$; see Eq. 7). If instead,*

$$\lim_{\Delta t \to 0} \frac{\lambda(\Delta t)}{\Delta t} = 0 \ , \tag{48}$$

*then the covariance of $\boldsymbol{J}$ is also finite.*

*As a consequence of the above, in the limit as $N \to \infty$ and $\Delta t \to 0$ with $N\Delta t = c$ held constant,*

$$\lim_{\Delta t \to 0, N \to \infty} Cov[\hat{s}_i(\boldsymbol{x}_t, t), \hat{s}_j(\boldsymbol{x}_{t'}, t')] = \frac{\delta_{ij}}{g_0^2 Z_\sigma c} \boldsymbol{\phi}(\boldsymbol{x}_t, t)^T \bar{\boldsymbol{K}}^{-1} \bar{\boldsymbol{K}}(0) \bar{\boldsymbol{K}}^{-1} \boldsymbol{\phi}(\boldsymbol{x}_{t'}, t') \tag{49}$$

*where the expectation is taken over sample realizations.*

By a completely analogous argument (which we do not explicitly write out here), we have, e.g.,

$$\mathbb{E}_{t,\boldsymbol{x}_0,\boldsymbol{x}_t|\{t,\boldsymbol{x}_0\}}(J_{ij} J_{k\ell} J_{ab}) \propto \mathbb{E}_{t,\boldsymbol{x}_0,\boldsymbol{x}_t|\{t,\boldsymbol{x}_0\}}\Big(\frac{s_j s_\ell s_b}{\sigma_t^3}\Big) \propto \frac{1}{(\Delta t)^{3/2}}(\delta_{j\ell} + \delta_{jb} + \delta_{\ell b})$$

$$\mathbb{E}_{t,\boldsymbol{x}_0,\boldsymbol{x}_t|\{t,\boldsymbol{x}_0\}}(J_{ij} J_{k\ell} J_{ab} J_{cd}) \propto \mathbb{E}_{t,\boldsymbol{x}_0,\boldsymbol{x}_t|\{t,\boldsymbol{x}_0\}}\Big(\frac{s_j s_\ell s_b s_d}{\sigma_t^4}\Big) \propto \frac{1}{(\Delta t)^3}(\delta_{j\ell}\delta_{bd} + \delta_{jb}\delta_{\ell d} + \delta_{jd}\delta_{\ell b}) \ .$$

Similar results hold for the higher-order moments—the expectation of a product of $2n$ $\boldsymbol{J}$ matrix elements goes like $(\Delta t)^{-2n+1}$, and the expectation of a product of $2n+1$ $\boldsymbol{J}$ matrix elements goes like $(\Delta t)^{-2n+1/2}$. (This has to do with the requirement that the score targets be paired in order to produce singular behavior.) This matters because, by the central limit theorem, the $n$-th order moment of the estimator $\hat{\boldsymbol{J}}$ scales like $N^{-n+1}$. Hence, third-order and higher moments scale like

$$\frac{1}{N^{2n-1}(\Delta t)^{2n-1}} = \frac{1}{c^{2n-1}} \ ( \ k = 2n \ )$$
$$\frac{1}{N^{2n}(\Delta t)^{2n-1/2}} \to 0 \quad ( \ k = 2n+1 \ ) \ . \tag{50}$$

As long as $c$ is somewhat larger than 1, it is reasonable to neglect these higher-order terms.

### C.3 EFFECTIVE REVERSE DIFFUSION SDE

The form of the V kernel follows directly from Proposition 3 and the path integral representation of the probability flow ODE (Eq. 11). What remains is to convert the path integral representation into a statement about the effective reverse diffusion dynamics. This is straightforward to those familiar with the correspondence between path integrals, SDEs, and the Fokker-Planck equation, but we spell out the relationship here via an elementary argument for the reader's convenience.

**Proposition 4 (Effective reverse diffusion SDE)** *The path integral representation of reverse diffusion dynamics averaged over sample realizations, i.e.,*

$$\mathbb{E}[q(\boldsymbol{x}_0|\boldsymbol{x}_T, \boldsymbol{\theta})] = \int \mathcal{D}[\boldsymbol{p}(t)]\mathcal{D}[\boldsymbol{x}(t)] \exp\{M_1 + M_2 + \cdots\}$$

$$M_1 := \int_0^{t_{max}} i\boldsymbol{p}(t) \cdot \left[\dot{\boldsymbol{x}}(t) + \beta_t \boldsymbol{x}_t + \frac{1}{2}g_t^2 \boldsymbol{s}_*(\boldsymbol{x}_t, t)\right] dt$$

$$M_2 := -\frac{1}{2}\int_0^{t_{max}} \int_0^{t_{max}} \boldsymbol{p}(t)^T \boldsymbol{V}(\boldsymbol{x}_t, t, \boldsymbol{x}_{t'}, t')\boldsymbol{p}(t')dtdt' \, ,$$

*where the V kernel $\boldsymbol{V}$ is as in Theorem 1, is equivalent to the distribution obtained by integrating the backwards-time SDE*

$$\dot{\boldsymbol{x}}_t = -\beta_t \boldsymbol{x}_t - \frac{1}{2}g_t^2 \boldsymbol{s}_*(\boldsymbol{x}_t, t) + \boldsymbol{\xi}(\boldsymbol{x}_t, t) \tag{51}$$

*from $t = t_{max}$ to $t = 0$ with initial condition $\boldsymbol{x}(t_{max}) = \boldsymbol{x}_T$. The noise term $\boldsymbol{\xi}(\boldsymbol{x}_t, t)$ is generically correlated across different times, and has*

$$Cov_{t,t',\boldsymbol{x}_t|t,\boldsymbol{x}_{t'}|t'}[\xi_i(\boldsymbol{x}_t, t), \xi_j(\boldsymbol{x}_{t'}, t')] = V_{ij}(\boldsymbol{x}_t, t, \boldsymbol{x}_{t'}, t') \, . \tag{52}$$