# OpenReview forum: "High variance score function estimates help diffusion models generalize"
_ICLR.cc/2024/Conference — Submitted to ICLR 2024_

### Official Review · Reviewer_dwfL · 2023-10-29

**Soundness:** 2 fair
**Presentation:** 2 fair
**Contribution:** 2 fair
**Rating:** 5
**Confidence:** 4

**Summary:**

The papers attempts the answer the question why diffusion models generalize well beyond the training dataset. To address this question, the paper studies linear score estimator and derives the optimal solution. Furthermore, the authors study the covariance of the optimal parameters. Some examples are given to illustrate this idea.

**Strengths:**

1. The authors study the linear score estimator class. The closed-form optimal solution to DSM is obtained.
2. The paper further investigates the covariance of the parameters in the score estimator. The authors find that the phenomenon of high variance.

**Weaknesses:**

1. The mathematical derivation in the paper looks like heuristics instead of rigorous proof. It is hard to tell the correctness of the arguments.
2. No experiments are provided to justify the high variance arguments.
3. Although the starting point of the paper is on the generalization of diffusion models, it is unclear how the high variance of the score estimators helps model generalization.
4. The writing can be improved.

**Questions:**

1. The paper only considers the linear score estimator and derives the optimal closed-form solution. How do you know the ground truth score function is linear? For a general score function, can we still have high variance parameters?
2. The most important issue with this paper is that I cannot tell how the high variance is connected to the generalization of diffusion models.
3. Can you provide some numerical experiments to support your claims in the paper?

---

> ### Author Response · Authors · 2023-11-22
>
> Thanks for reading the paper and offering comments. Hopefully the new version of the paper (to be posted before AOE Nov 22) addresses your concerns. See below for detailed responses:
>
> **Weaknesses:**
>
> > 1. The mathematical derivation in the paper looks like heuristics instead of rigorous proof. It is hard to tell the correctness of the arguments.
>
> We agree that this was a weakness of the original version, and the mathematical content has been substantially reorganized (and made more rigorous) to fit a more standard theorem/proof structure. The original result included in the paper has also been substantially generalized to remove two hypotheses; the result no longer requires a gaussian mixture training distribution or that score function parameters are learned separately for different times.
>
> > 2. No experiments are provided to justify the high variance arguments.
>
> New experiments have been done to justify the high variance arguments (although we still refer heavily to previous work). Two important pieces of empirical evidence are that (i) the score function learned by models is usually not the same as the "empirical" score of the data set, and (ii) models trained using what we call the "naive" objective exhibit more memorization.
>
> > 3. Although the starting point of the paper is on the generalization of diffusion models, it is unclear how the high variance of the score estimators helps model generalization.
>
> We agree that this could have been better explained in the original version. Two new sections have been added to the newest version to remedy this: one presenting some intuition about generalization, and another discussing concrete examples of how the additional variance discussed in the text implements an inductive bias that encourages interpolation (filling in gaps between training data), extrapolation (extending patterns in the training data), and feature blending (generating samples which include both feature $X$ and feature $Y$ even when training examples only involve one of the two features).
>
> > 4. The writing can be improved.
>
> The aforementioned new sections have been added, and the discussion was clarified and reorganized in various places in order to try to remedy this issue.
>
> **Questions:**
>
> > 1. The paper only considers the linear score estimator and derives the optimal closed-form solution. How do you know the ground truth score function is linear? For a general score function, can we still have high variance parameters?
>
> It is important to note that the score estimators considered in this paper are linear in *features*, not linear in *states*. This means that a variety of arbitrarily expressive function approximators (e.g., a Fourier basis, Gaussian basis functions, orthogonal polynomials) are permitted. This point of confusion has been clarified in the text.
>
> Another case of interest is that of neural networks in the neural tangent kernel (NTK) regime. These are effectively linear, and the main result of the paper holds for them also. (A brief discussion of this has been added to the paper.) So additional effective variance due to noisy parameter estimates is a phenomenon that appears to happen for a fairly large class of models, although for reasons of mathematical tractability we are unable to discuss all possible models (e.g., neural networks in the "rich" learning regime) here.
>
> > 2. The most important issue with this paper is that I cannot tell how the high variance is connected to the generalization of diffusion models.
>
> See above answer. Two new sections have been added to discuss this explicitly.
>
> > 3. Can you provide some numerical experiments to support your claims in the paper?
>
> Various numerical experiments have been added in order to both (i) convince the reader that the high variance phenomenon is real, and (ii) exhibit the inductive biases it produces.

---

> > ### Comment · Reviewer_dwfL · 2023-11-23
> > **Response to the rebuttal**
> >
> > Thanks for your response. I appreciate your efforts on the revision. Most of my questions are well-addressed. However, I still have some minor comments:
> >
> > 1. It is fine to argue that the feature map is nonlinear in the state but the NTK should not be covered. Although NTK is essentially a linear model, its mathematical form differs from the classical random feature models.
> > 2. It would be better to highlight your main changes in the revision. It's still hard to tell your revision in the updated manuscript.
> >
> > Based on this situation, I do believe my original evaluation is fair.

---

### Official Review · Reviewer_TEfw · 2023-10-31

**Soundness:** 2 fair
**Presentation:** 3 good
**Contribution:** 3 good
**Rating:** 5
**Confidence:** 4

**Summary:**

This paper studies the effect of high variance of of score function estimates at early times. The key idea is to identify the score estimates by running the backward diffusion with the optimal score and convolving with some kernel function -- in other words diffusion models are similar to some kernel density estimation.

**Strengths:**

This paper tries to explain mathematically the effect of high variance of score estimation at early times in diffusion models -- which was observed empirically. The key takeaway is to identify the diffusion training by kernel density problems. This connection appears to be novel and insightful. The paper is generally well written, and I mostly enjoyed reading it. I have checked most computations, and they seem to be correct.

**Weaknesses:**

The main concern is that the paper uses some simple diffusion models (1)-(2), as well as the Gaussian mixture training distribution. This limits the applicability of the results. For instance, there are more advanced diffusion models, e.g. VE, VP, sub-VP... What happens in these cases? Is it possible to identify all theses diffusion models by a suitable kernel density problem? The authors may want to comment or explain.

Also it is not clear whether the "theoretical" computations can carry over to other distributions than Gaussian mixture (or a mixture of delta mass). The authors may want to comment on this.

**Questions:**

See the weaknesses.

---

> ### Author Response · Authors · 2023-11-22
>
> Thanks for reading the paper and offering comments. Hopefully the new version of the paper (to be posted before AOE Nov 22) addresses your concerns. See below for detailed responses:
>
> **Weaknesses:**
>
> > The main concern is that the paper uses some simple diffusion models (1)-(2), as well as the Gaussian mixture training distribution. This limits the applicability of the results. For instance, there are more advanced diffusion models, e.g. VE, VP, sub-VP... What happens in these cases? Is it possible to identify all theses diffusion models by a suitable kernel density problem? The authors may want to comment or explain.
>
> A substantial amount of effort has been put into generalizing the results. In the newest version, two assumptions are no longer required: the training distribution need not be a Gaussian mixture, and score estimator parameters need not be learned independently for different times.
>
> The score estimator is still assumed to be linear in a set of feature maps (which are now allowed to be both state- and time-dependent), but this is actually a fairly general class of estimators and includes many arbitrarily expressive function approximators (e.g., a Fourier basis, Gaussian basis functions, orthogonal polynomials). Importantly, the key assumption is that the score is linear in these features, not linear in *states*. This is a point of confusion that has been clarified in the text.
>
> One case which is somewhat realistic that is covered now is the case of neural networks in the neural tangent kernel (NTK) regime. Because these are (to very good approximation, in the infinite width limit) linear in their parameters, they are now covered by the main result of the newest version. This takes the main result closer to SOTA models, although obviously there is a ways to go (e.g., what about neural networks in the so-called "rich" learning regime?). We still think this is a substantial theoretical contribution that could serve as a foundation for future work to generalize in various ways.
>
> VP, VE, and sub-VP models are all covered by the main result, and some discussion of their differences has been added to the main text. We agree that this is a practically relevant case worthy of explicit discussion.
>
> Yes, all cases can be identified with a suitable kernel density problem. One nuance is that, in the new (more general) result, where feature maps are allowed to be time-dependent, the result generically includes an effective noise term in the probability flow ODE (as opposed to just a convolution at the end; see below).
>
>
> > Also it is not clear whether the "theoretical" computations can carry over to other distributions than Gaussian mixture (or a mixture of delta mass). The authors may want to comment on this.
>
> This assumption has been removed in the newest version, and turns out to be completely unimportant.
>
> ------------------
>
> For completeness' sake, here is the main theorem in the newest version:
>
> **Theorem 1.** [**Linear score estimators trained via DSM asymptotically generalize**] Suppose that the parameters of a linear score estimator are optimized according to the DSM objective using $N$ independent samples from $\lambda_0(t) p(\mathbf{x}_0) p(\mathbf{x}_t | \mathbf{x}_0, t)$. Consider the result of reverse diffusion using this estimator by Euler-integrating the probability flow ODE with a small time step $\Delta t$. If $N \to \infty$ and $\Delta t \to 0$ with $N \Delta t = c$ held constant, then sampling from $\mathbb{E}[ q(\mathbf{x}_0 | \mathbf{x}_T, \boldsymbol{\theta}) ]$ is equivalent to simulating the backwards-time (Ito-interpreted) SDE
>
> $$\dot{\mathbf{x}}_ t = -\beta_ t \mathbf{x}_ t - \frac{1}{2} {g_ t}^2 \mathbf{s}_ *(\mathbf{x}_ t, t) + \boldsymbol{\xi}(\mathbf{x}_ t, t) $$
>
> from $t = t_{max}$ to $t = 0$ with initial condition $\mathbf{x}(t_{max}) = \mathbf{x}_T$. The noise term $\boldsymbol{\xi}(\mathbf{x}_t, t)$ is generically correlated across different times, and has
>
> $$\text{Cov}_ {t, t', \mathbf{x}_ t | t, \mathbf{x}_ {t'} | t'}[ {\xi}_ {i} (\mathbf{x}_ t, t), {\xi}_ {j} (\mathbf{x}_ {t'}, t') ] = V_ {ij} (\mathbf{x}_ {t}, \mathbf{x}_ {t'}, t, t')$$
>
> where we define the $D \times D$  "V kernel" $\mathbf{V}$ via
> $$
> V_ {ij} := \frac{\delta_ {ij}}{{g_ 0}^2 Z_ {\sigma} c} \  \left( \frac{{g_ {t}}^2}{2 \sigma_ t} \cdot \frac{{g_ {t'}}^2}{2 \sigma_ {t'}} \right)  \ \boldsymbol{\phi}(\mathbf{x}_ t, t)^T \bar{\mathbf{K}}^{-1} \bar{\mathbf{K}}(0) \bar{\mathbf{K}}^{-1} \boldsymbol{\phi}(\mathbf{x}_ {t'}, t') .
> $$

---

### Official Review · Reviewer_oZCe · 2023-10-31

**Soundness:** 1 poor
**Presentation:** 2 fair
**Contribution:** 1 poor
**Rating:** 3
**Confidence:** 4

**Summary:**

This paper proposes to study the generalization ability of diffusion models: why diffusion models do not simply remember the training set but can generate new samples. The paper argues that a key factor is the high variance score function estimates at small $t$. The paper studies the behavior of diffusion models with a specific setting where they are parametrized as a linear function of features, and shows that they learn a distribution that is mathematically equivalent to convolving the optimal distribution with a particular kernel function.

**Strengths:**

The issue the paper tries to study is an important one. While diffusion models are being deployed quite literally everywhere, it is paramount that we understand what makes them generalize.

**Weaknesses:**

1. My biggest complaint about the paper is that I am not convinced by the paper's argument of the high variance score function estimates at small $t$, which also serves as the paper's motivation.
    - I am not sure why in Eq.6, we are looking at the covariance over $x_t$, since the network is current-position-aware (the network has $x_t$ and $t$ as input). The network prediction targets are often regularized to have a unit norm [1], so the network predicted scale stays the same across $t$, which is not considered in the paper. Also, people have tried to directly predict $x$ through a different parameterization, which will make the prediction targets very stable at small $t$. This would invalidate the paper's analysis, and we still do not see perfect memorization happen.
    - It makes more sense to consider the variance explored in [2], which is over $p_{x_0|x_t}$. It is also observed there that the variance of score function estimates is actually small with small $t$. In fact, in [2], it is observed that explicitly minimizing the variance of the score function estimation often leads to better results.
    - The denoised score matching objective is always trying to fit the injected random noise. The ground truth score (marginalized over all samples) is very different only with moderate t. I agree that fitting small $t$ score functions is difficult as mentioned in prior works, but that is a different issue.
2. The paper's analysis heavily relies on a particular parameterization (not used in practice), which in my opinion, is the cause of the observed phenomenon, instead of the denoising score matching loss as the authors suggest. What if the model can actually learn the distribution perfectly? Will the analysis in this paper still hold? Equivalently, if the network is trained with the "naive" approach, does the model actually learn the distribution perfectly? (with all the training details corrected, like sampling distribution of $\sigma$ and network prediction target normalization) If not, then I am not sure how much of a contribution this paper is to the field, as the particular setting considered in this paper is impractical, and does not translate very well to the real usage of diffusion models.
3. In the introduction, the paper rules out many candidates for generalization, purely based on prior works, intuitions, and observations. In order to make these claims, careful study is required. For example, the authors mention that modern architectures are flexible enough to in principle learn the optimal score. The authors need to back up the claims carefully, because that implies that you can perfectly reconstruct the dataset with these networks, which does not happen.
4. Since the issue considered in this paper is really an empirical one (because the optimal solution to diffusion model training is memorizing all training samples), I highly suggest the authors do some experiments. Related issue: the presentation can be significantly improved with figures. The authors should have more than enough space in the current version.

[1] Karras et al. "Elucidating the design space of diffusion-based generative models." NeurIPS 2022.

[2] Xu et al. "Stable target field for reduced variance score estimation in diffusion models." ICLR 2023.

**Questions:**

1. Where is Eq.8 used in practice? Or, equivalently, which diffusion models used in practice actually underweight small $t$ during training? I see the authors also cite [1], which used a log-normal distribution, and they basically train with more samples at small $t$ compared to large $t$. This fact directly counters one of the assumptions listed in the last paragraph in Sec.3: “the choice of a time sampling distribution that underweights small times…”

[1] Karras et al. "Elucidating the design space of diffusion-based generative models." NeurIPS 2022.

---

> ### Author Response · Authors · 2023-11-22
>
> Thanks for reading the paper and offering comments. Hopefully the new version of the paper (to be posted before AOE Nov 22) addresses your concerns. See below for detailed responses:
>
> **Weaknesses**
>
> > My biggest complaint about the paper is that I am not convinced by the paper's argument of the high variance score function estimates at small $t$, which also serves as the paper's motivation. I am not sure why in Eq.6, we are looking at the covariance over
> $x_t$, since the network is current-position-aware (the network has and $x_t$ and $t$ as input).
>
> This brief discussion is just meant to provide some intuition about a crucial difference between what we are calling the "naive" objective and the DSM objective. What actually matters is the covariance of the target as a function of all variables, i.e., with respect to $p(\mathbf{x}_ t | \mathbf{x}_ 0, t) p(\mathbf{x}_ 0) \lambda(t)$. Integrating wrt $x_ 0$ doesn't change anything, and integrating with respect to $t$ (assuming $\lambda(t) \sim \sigma_ t^2$) produces the integral of a constant, and hence proportional to $t_{max}$.
>
> Other authors (like Song et al. 2021) have explicitly considered the covariance of the target with respect to $\mathbf{x}_t$ alone, so I think it's a reasonable thing to look at. Moreover, in the (now more general) central result of the paper, this variance is indeed the key ingredient that creates what we are calling a type of generalization.
>
> > The network prediction targets are often regularized to have a unit norm [1], so the network predicted scale stays the same across $t$, which is not considered in the paper. Also, people have tried to directly predict $x$
>  through a different parameterization, which will make the prediction targets very stable at small $t$. This would invalidate the paper's analysis, and we still do not see perfect memorization happen.
>
> I think this is a misunderstanding, although it is a subtle and important one. The situation is somewhat confusing because different authors use different notation, sometimes using the same symbol to mean very different things. A new appendix has been added that discusses this particular point, since it is potentially a major source of confusion.
>
> Adding to the confusion was the previous choice of learning parameters separately for different times. In the newest version, this restriction has been removed, so feature maps are allowed to depend on both state and time in a fairly arbitrary way (with some restriction, like square-integrability with respect to the relevant distributions, and smoothness).
>
> The situation considered in the current version of the paper is (we will argue) very similar to current SOTA models. The time-sampling distribution used in the main result is
> $$ \lambda_ *(t) = \frac{\sigma_ t^2}{\int_0^{t_ {max}} \sigma_ t^2 \ dt} = \frac{\sigma_ t^2}{Z_ {\sigma}} .$$
> In the newest version of the paper, we are explicit about scaling the model as $\mathbf{\epsilon}_ {\boldsymbol{\theta}}(\mathbf{x}_ t, t) = \sigma_ t \hat{\mathbf{s}}_ {\mathbf{\theta}}(\mathbf{x}_ t, t)$ (i.e., the function approximator $\mathbf{\epsilon}$ learns the true score multiplied by $\sigma_ t$). With this change, the objective becomes
> $$
> J_{1}(\boldsymbol{\theta}) = \frac{1}{2 Z_ {\sigma}} \int \ \Vert \boldsymbol{\epsilon}_ {\boldsymbol{\theta}}(\alpha_ t \mathbf{x}_ 0 + \sigma_ t \boldsymbol{\epsilon}, t) - \boldsymbol{\epsilon} \Vert_2^2 \ \mathcal{N}(\boldsymbol{\epsilon}; \mathbf{0}, \mathbf{I}) p(\mathbf{x}_ 0) \ d\boldsymbol{\epsilon} d\mathbf{x}_ 0 dt   $$
> This is exactly the objective used by, e.g., Stable Diffusion (Rombach et al 2022; see Eq. 1) and the PNDM paper (Ho et al 2020; see Eq. 14). It turns out that the way we handled the independent-time case hid this scaling (although it was still implicitly there). Some discussion of the variance normalization is now present in the revised version.

---

> > ### Author Response · Authors · 2023-11-23
> >
> > > It makes more sense to consider the variance explored in [2], which is over $p_{x_0 | x_t}$. It is also observed there that the variance of score function estimates is actually small with small $t$. In fact, in [2], it is observed that explicitly minimizing the variance of the score function estimation often leads to better results.
> >
> > Thanks for the pointer, this paper is cited in the revised version of the paper. While $p_{x_0 | x_t}$ is indeed the natural distribution to consider for, e.g., $E[ \nabla_ {\mathbf{x}_ t} \log p(\mathbf{x}_ t | \mathbf{x}_ 0) ]$, we kind of disagree that there is a problem with considering it here. Again, that statement is just for intuition. Additionally, we believe that this intuition is valuable, since the singular estimate behavior appears to be a direct consequence of this (see details of new math).
> >
> > While the Xu et al. paper is both important and interesting, the regime of concern is somewhat different than for our paper. In particular, Xu et al. focus on reducing variance in the so-called "intermediate" regime (moderate $t$ / noise scale), where different modes in the data compete with one another. The generalization behavior we established mathematically relates to the small $t$ regime (what Xu et al call "Phase 1"). The generalization behavior is in fact *entirely* due to the time sampling distribution locally undersampling at small times $t$.
> >
> > It is probably true given Xu et al's work that variance reduction is useful in the intermediate regime, but this is an orthogonal claim. It is almost tautologically true, given a finite set of examples (and hence a training distribution that is a mixture of delta functions), that we want *additional* variance for very small noise scales, since otherwise we would simply be attracted to one of the training example modes, and hence just reproduce a training example. Some explicit discussion of this has been added to the new Section 3, about the intuition for our claims.
> >
> > > The paper's analysis heavily relies on a particular parameterization (not used in practice), which in my opinion, is the cause of the observed phenomenon, instead of the denoising score matching loss as the authors suggest. What if the model can actually learn the distribution perfectly? Will the analysis in this paper still hold?
> >
> > In the revised version of the paper, the main result has been substantially generalized, with (i) correct variance normalization made explicit, and (ii) the separate-times assumption relaxed. In the current setting, the score estimator is assumed to be linear in some set of (generally state- and time-dependent) feature maps.
> >
> > Although this does not cover all possible function approximators, it is still a fairly broad class. Included are arbitrarily expressive approximators (e.g., a Fourier basis, Gaussian basis functions, orthogonal polynomials) for which there is no expressivity-related reason that the "true" score function cannot be learned. Also included in this class are neural networks in the neural tangent kernel (NTK) regime, since they are effectively linear in their learned parameters. A brief discussion of this specific, fairly interesting example has been added to the paper. The analysis of the paper still holds for such approximators.
> >
> > > Equivalently, if the network is trained with the "naive" approach, does the model actually learn the distribution perfectly? (with all the training details corrected, like sampling distribution of $\sigma$ and network prediction target normalization) If not, then I am not sure how much of a contribution this paper is to the field, as the particular setting considered in this paper is impractical, and does not translate very well to the real usage of diffusion models.
> >
> > *Yes*, even an arbitrarily expressive function approximator trained with the naive objective instead of the DSM one exactly reproduces the "true"/empirical score, that of a mixture of delta functions for example. A result saying as much is explicitly incorporated in the newest version of the paper. We believe this is an important theoretical result, since it establishes rigorously that the DSM objective is useful for reasons of more than just convenience: it aids generalization.
> >
> > One last comment here: while our result does not cover all possible function approximators, we believe that the class balances practically interesting (since it includes arbitrarily expressive approximators) with mathematically tractable. Doing theory on diffusion models is currently hard and understanding is limited; it would be difficult to generalize to more realistic cases, and makes sense to rigorously establish results in simpler settings first.

---

> > > ### Author Response · Authors · 2023-11-23
> > >
> > > > In the introduction, the paper rules out many candidates for generalization, purely based on prior works, intuitions, and observations. In order to make these claims, careful study is required. For example, the authors mention that modern architectures are flexible enough to in principle learn the optimal score. The authors need to back up the claims carefully, because that implies that you can perfectly reconstruct the dataset with these networks, which does not happen.
> > >
> > > We have softened our claims here, since their precise form is just for intuition. However, we maintain that modern architectures are indeed flexible enough to in principle learn the optimal score. What we mean is that approximators like neural networks can approximate arbitrary functions; this is just mathematically true. Indeed, it is empirically true that real diffusion models do not perfectly reconstruct the data set. All we are saying is that this cannot be due to a lack of expressivity on the part of the function approximator.
> > >
> > > Our new theoretical result also covers some related cases, as mentioned above. For example, a fully-connected ReLu network with one hidden layer in the infinite width limit is arbitrarily expressive, and is covered by our main theorem. If trained using the naive objective, we prove theoretically that it *does* perfectly reconstruct the data set.
> > >
> > > > Since the issue considered in this paper is really an empirical one (because the optimal solution to diffusion model training is memorizing all training samples), I highly suggest the authors do some experiments. Related issue: the presentation can be significantly improved with figures. The authors should have more than enough space in the current version.
> > >
> > > We have added some small experiments to the paper, as well as a few small figures. However, it is worth noting that what is probably lacking in the diffusion field right now is not more experiments, but a deeper theoretical understanding of why these models generalize. In this sense, the issue cannot totally be resolved via numerical experiments.
> > >
> > > **Questions**
> > >
> > > > Where is Eq.8 used in practice? Or, equivalently, which diffusion models used in practice actually underweight small
> > >  during training? I see the authors also cite [1], which used a log-normal distribution, and they basically train with more samples at small $t$ compared to large $t$. This fact directly counters one of the assumptions listed in the last paragraph in Sec.3: “the choice of a time sampling distribution that underweights small times…”
> > >
> > > You appear to have this backwards. In the revised version, we have made the time-dependence of the DSM objective explicit since we are no longer forcing ourselves to learn the score at different times separately:
> > > $$J_ 1 = \frac{1}{2} \int \lambda(t) \ \Vert \hat{\mathbf{s}}_ {\boldsymbol{\theta}}(\mathbf{x}_ t, t) - \nabla_ {\mathbf{x}_ t} \log p(\mathbf{x}_ t | \mathbf{x}_ 0) \Vert_ 2^2 \ p(\mathbf{x}_ t | \mathbf{x}_ 0) p(\mathbf{x}_ 0) \ d\mathbf{x}_ t d\mathbf{x}_ 0 dt$$
> > >
> > > The standard choice (used by Song et al 2021, PNDM, Stable Diffusion...) is to take $\lambda(t) \propto \sigma_t^2$, and this is the choice we consider in this paper. Since $\sigma_t^2 \to 0$ for small $t$ (and in particular, $\sigma_t^2 \propto \Delta t$ for small $t$ in all cases we consider, including VP, VE, sub-VP...the kinds of models defined by Eq 1-3), small times are by construction undersampled.
> > >
> > > The EDM paper identifies time and noise scale, so $\sigma_t = t$. They separately consider the distribution $p_{train}(\sigma)$ of noise scales and an effective weighting factor $\lambda(\sigma) c_ {out}(\sigma)^2$ (see near Eq. 8); these two things can be combined into one effective noise sampling distribution. We have
> > > $$p_{eff}(\sigma) = p_{train}(\sigma) \lambda(\sigma) c_{out}(\sigma)^2  .$$
> > > They recommend $\lambda(\sigma) \sim  c_{out}(\sigma)^2$ (pg 9), and $p_{train}(\sigma)$ to be log-normal. Then
> > > $$p_{eff}(\sigma) = p_{train}(\sigma)$$
> > > is log-normal. Log-normal distributions actually approach zero fairly quickly, although not as quickly as $t$---hence, small times / noise scales are indeed undersampled, quite drastically. See Figure 5A from the EDM paper for some illustration of this. Note also, in the same panel, that the loss for the score function at small times is somewhat large after training (partly as a consequence of this undersampling!). Not only does this not counter our assumption, it is in fact evidence in our favor. Maybe the confusion comes from the meaning of "small"; here, we really mean 'close to zero', rather than intermediate times / noise scales. We have tried to make this clearer in the revision.

---

> ### Comment · Reviewer_oZCe · 2023-12-02
>
> I thank the authors for their efforts in revising the paper. I highly suggest that the authors highlight their modifications in a different color, especially when the revision is of this significance. Since a lot of the suggested changes are not incorporated in the updated version, I cannot recommend acceptance. Nevertheless, I agree that understanding the generalization behavior of diffusion models is an important topic, and I hope to read the completed paper in the future.

---

### Official Review · Reviewer_V1yR · 2023-11-01

**Soundness:** 2 fair
**Presentation:** 3 good
**Contribution:** 2 fair
**Rating:** 3
**Confidence:** 3

**Summary:**

This paper investigates the following problem: How do diffusion-based generative modesl generalize beyond the training set? The authors claim that this phenomenon is at least partially explained by the fact that score function estimates have a large varaince. To support this claim, they consider a linear score estimator, where $\hat{s}_{\theta}(x_t, t) = W_0(t) + W(t) \phi(x_t). $ Here, $\phi$ is a fixed feature map, and $\theta(t) = (W_0(t), W(t))$ are the parameters to be estimated. To train the estimator, they consider the limit of large number of samples $N \to \infty$ and small times bins $\Delta t \to 0$. They also assume a specific time sampling distribution $\lambda(t)$. To learn $\theta(t_i)$, they use $N \lambda(t_i) \Delta t$ samples. They propose to estimate the kernel functions using a sample mean estimator. They show that the learned distribution will not be $q_{\ast}$, which is derived from the optimal linear estimator. Instead, they prove that the learned distribution will be $q_{\ast}$ convolved with a specific Gaussian kernel. They apply their results to several machine learning tasks.

**Strengths:**

This paper attempts to explain the generalization of diffusion models from a novel perspective: The high variance of the score function. Using a linear score function estimator, and assume an appropriate asymptotic regime for the training, they are able to explicitly characterize the distribution that is learned from data. This distribution is obtained with optimal score function convolved with a specific kernel. Their results find applications in many machine learning tasks and contribute to explain the generalization of diffusion models.

**Weaknesses:**

I have some doubts on the asymptotic regime considered in this paper. I feel it is not very efficient to use only data in a small time window to train the score function at that time point. As far as I am concerned, $\lambda(t)$ in the past papers was introduced to impose weights on the loss function instead of sample splitting. I think the authors should elaborate more on why the sample splitting scheme is a reasonable one. I feel in the main result they are getting variation because they do not have enough sample to train each single score function.

In addition, I feel using only linear score estimator is a bit restricted, as score function is defined as the gradient of the log density, I would expect that it is in general non-linear. Perhaps the authors can make their results more persuasive by giving several examples that have linear score functions?

**Questions:**

1. Is there a way to estimate the feature maps $\phi$ when they are not known a priori?
2. How accurately can a linear estimator learn the score function in typical situations? Maybe the authors can comment a little bit on that.
3. If we use a better estimator than taking the sample average to estimate the score function, do we get a better result? Will it hurt or imporve the generalization ability?
4. If a different sampling distribution is employed, how does the results change?
5. In practice, the number of samples used for training will be very large, hence $c$ would also be large. In this case, the variance according to the theorem will be small. Do we even expect generalization to happen in such a large-sample situation?

---

> ### Author Response · Authors · 2023-11-22
>
> Thanks for reading the paper and offering comments. Hopefully the new version of the paper (to be posted before AOE Nov 22) addresses your concerns. See below for detailed responses:
>
> **Weaknesses:**
>
> > I have some doubts on the asymptotic regime considered in this paper. I feel it is not very efficient to use only data in a small time window to train the score function at that time point. As far as I am concerned, $\lambda(t)$ in the past papers was introduced to impose weights on the loss function instead of sample splitting. I think the authors should elaborate more on why the sample splitting scheme is a reasonable one. I feel in the main result they are getting variation because they do not have enough sample to train each single score function.
>
> This is a good point, and it previously bothered us that we were only able to obtain a result for this (admittedly artificial) case. In the newest version, a substantial amount of work has been put in to generalize this result. Two restrictive hypotheses have been removed from the main result: sample splitting is no longer required (i.e., the feature maps are allowed to depend on time), and the training distribution need not be a Gaussian mixture. Even in the case of time-dependent feature maps, additional variance survives in the limit of many samples (as long as, as in the original result, the number of samples $N$ and the time step $\Delta t$ used to simulate the probability flow ODE scale in a comparable fashion). We think, with this change, the result is much more interesting.
>
> > In addition, I feel using only linear score estimator is a bit restricted, as score function is defined as the gradient of the log density, I would expect that it is in general non-linear. Perhaps the authors can make their results more persuasive by giving several examples that have linear score functions?
>
> This is a point of confusion other reviewers have had, so we have clarified it in the text. Our score function estimator need not be linear in *state* (i.e., $x_t$); it only need to be linear in some set of *feature maps*. Indeed, score functions are generally not linear in state (and the score function that *is* linear in state is that of a Gaussian).
>
> The class of score estimators linear in some set of feature maps is actually quite broad, and includes many arbitrarily expressive function approximators (e.g., a Fourier basis, Gaussian basis functions, orthogonal polynomials). An important example of an estimator which is linear in this sense is a neural network in the neural tangent kernel (NTK) regime, since it is effectively linear in its (learned) weights; a brief discussion of this example has been added to the paper.
>
> Some more explicit examples of different possible feature maps have been added in order to make this point clearer.
>
> > **Questions:**
>
> > 1. Is there a way to estimate the feature maps $\phi$ when they are not known a priori?
>
> We consider the problem of choosing 'good' feature maps out of the scope of this paper, since this is extremely similar to the question of which neural network architectures are best for diffusion models (and this is a difficult question for which the answer is not quite known, although it might involve transformers are convolutional neural nets). For our purposes, it suffices to use arbitrary expressive ones (e.g., an NTK neural net, or Fourier basis). Some examples of arbitrarily expressive feature maps have been added to the main text.
>
> > 2. How accurately can a linear estimator learn the score function in typical situations? Maybe the authors can comment a little bit on that.
>
> See above answers. A variety of arbitrarily expressive feature maps can be used. Some clarification related to this has been added to the text.
>
> > 3. If we use a better estimator than taking the sample average to estimate the score function, do we get a better result? Will it hurt or improve the generalization ability?
>
> Since sample averages are the provably optimal result of (full batch) gradient descent, we have decided to restrict the scope to this case. It is an interesting question for future work what happens when something different is used, e.g., gradient descent using mini-batches. A note of this has been added to the discussion.
>
>
> > 4. If a different sampling distribution is employed, how does the results change?
>
> Some discussion of different time sampling distributions $\lambda(t)$ (used during training) has been added to the main text. The particular choice of $\lambda(t)$ considered in this paper (and used in various SOTA models, see e.g. Song et al 2021 and Karras et al 2022) is special because of its small-time behavior; sampling distributions with $\lambda(dt) \approx (dt)^n$ for some $n > 1$ do not generalize in the sense described in this paper. Some discussion of this point has been added to the text.

---

> > ### Author Response · Authors · 2023-11-22
> >
> > > 5. In practice, the number of samples used for training will be very large, hence $c$ would also be large. In this case, the variance according to the theorem will be small. Do we even expect generalization to happen in such a large-sample situation?
> >
> > In practical situations, we would indeed expect generalization to happen for the following reason. We want to integrate the probability flow ODE accurately in order to sample from the learned distribution; this requires using small $\Delta t$ time steps. Heuristically, using smaller $\Delta t$ means we would like to learn the score function accurately for a larger number of values, which is possible when the number of samples $N$ becomes larger. Theoretically, we want $N$ and $\Delta t$ to a scale in a comparable fashion (i.e., $N \Delta t$ = constant) in order for the model to learn something other than the optimal score. Intuitively, what happens is that you get more and more information to learn an ever larger number of values, so it can never be done completely perfectly.
> >
> > In practice, $N$ is not infinite and $\Delta t$ is not zero. However, usually $N$ is somewhat large and $\Delta t$ is somewhat small, making this limit not totally unreasonable. The situation is somewhat similar to the infinite width limit of neural networks: a width does not have to be *that* large in order to resemble an 'infinitely' wide network.
> >
> > ----------
> >
> >
> > For completeness' sake, here is the main theorem in the newest version. It involves no sample splitting in general, getting rid of the previous major weakness.
> >
> > **Theorem 1.** [**Linear score estimators trained via DSM asymptotically generalize**] Suppose that the parameters of a linear score estimator are optimized according to the DSM objective using $N$ independent samples from $\lambda_0(t) p(\mathbf{x}_0) p(\mathbf{x}_t | \mathbf{x}_0, t)$. Consider the result of reverse diffusion using this estimator by Euler-integrating the probability flow ODE with a small time step $\Delta t$. If $N \to \infty$ and $\Delta t \to 0$ with $N \Delta t = c$ held constant, then sampling from $\mathbb{E}[ q(\mathbf{x}_0 | \mathbf{x}_T, \boldsymbol{\theta}) ]$ is equivalent to simulating the backwards-time (Ito-interpreted) SDE
> >
> > $$\dot{\mathbf{x}}_ t = -\beta_ t \mathbf{x}_ t - \frac{1}{2} {g_ t}^2 \mathbf{s}_ *(\mathbf{x}_ t, t) + \boldsymbol{\xi}(\mathbf{x}_ t, t) $$
> >
> > from $t = t_{max}$ to $t = 0$ with initial condition $\mathbf{x}(t_{max}) = \mathbf{x}_T$. The noise term $\boldsymbol{\xi}(\mathbf{x}_t, t)$ is generically correlated across different times, and has
> >
> > $$\text{Cov}_ {t, t', \mathbf{x}_ t | t, \mathbf{x}_ {t'} | t'}[ {\xi}_ {i} (\mathbf{x}_ t, t), {\xi}_ {j} (\mathbf{x}_ {t'}, t') ] = V_ {ij} (\mathbf{x}_ {t}, \mathbf{x}_ {t'}, t, t')$$
> >
> > where we define the $D \times D$  "V kernel" $\mathbf{V}$ via
> > $$
> > V_ {ij} := \frac{\delta_ {ij}}{{g_ 0}^2 Z_ {\sigma} c} \  \left( \frac{{g_ {t}}^2}{2 \sigma_ t} \cdot \frac{{g_ {t'}}^2}{2 \sigma_ {t'}} \right)  \ \boldsymbol{\phi}(\mathbf{x}_ t, t)^T \bar{\mathbf{K}}^{-1} \bar{\mathbf{K}}(0) \bar{\mathbf{K}}^{-1} \boldsymbol{\phi}(\mathbf{x}_ {t'}, t') .
> > $$

---

### Author Response · Authors · 2023-11-23
**Summary of changes**

Thanks to the reviewers for their helpful comments. I was unhappy with some of the choices in the original draft (like the assumption that different score function parameters are learned at different times), and so have spent most of the discussion period trying to generalize these choices. Although I have been somewhat successful in doing so (see the revised draft), I unfortunately ran out of time and have not had the opportunity to implement many of the suggested/proposed/planned changes. So the current state of things is that there is an interesting and highly nontrivial generalization-related result that should be of general interest, but it is not presented very well (e.g., there are still no figures). Sorry about this, but I hope you nonetheless have found some of this interesting!

Here is the new main Theorem, for reference:

**Theorem 1.** [**Linear score estimators trained via DSM asymptotically generalize**] Suppose that the parameters of a linear score estimator are optimized according to the DSM objective using $N$ independent samples from $\lambda_0(t) p(\mathbf{x}_0) p(\mathbf{x}_t | \mathbf{x}_0, t)$. Consider the result of reverse diffusion using this estimator by Euler-integrating the probability flow ODE with a small time step $\Delta t$. If $N \to \infty$ and $\Delta t \to 0$ with $N \Delta t = c$ held constant, then sampling from $\mathbb{E}[ q(\mathbf{x}_0 | \mathbf{x}_T, \boldsymbol{\theta}) ]$ is equivalent to simulating the backwards-time (Ito-interpreted) SDE

$$\dot{\mathbf{x}}_ t = -\beta_ t \mathbf{x}_ t - \frac{1}{2} {g_ t}^2 \mathbf{s}_ *(\mathbf{x}_ t, t) + \boldsymbol{\xi}(\mathbf{x}_ t, t) $$

from $t = t_{max}$ to $t = 0$ with initial condition $\mathbf{x}(t_{max}) = \mathbf{x}_T$. The noise term $\boldsymbol{\xi}(\mathbf{x}_t, t)$ is generically correlated across different times, and has

$$\text{Cov}_ {t, t', \mathbf{x}_ t | t, \mathbf{x}_ {t'} | t'}[ {\xi}_ {i} (\mathbf{x}_ t, t), {\xi}_ {j} (\mathbf{x}_ {t'}, t') ] = V_ {ij} (\mathbf{x}_ {t}, \mathbf{x}_ {t'}, t, t')$$

where we define the $D \times D$  "V kernel" $\mathbf{V}$ via
$$
V_ {ij} := \frac{\delta_ {ij}}{{g_ 0}^2 Z_ {\sigma} c} \  \left( \frac{{g_ {t}}^2}{2 \sigma_ t} \cdot \frac{{g_ {t'}}^2}{2 \sigma_ {t'}} \right)  \ \boldsymbol{\phi}(\mathbf{x}_ t, t)^T \bar{\mathbf{K}}^{-1} \bar{\mathbf{K}}(0) \bar{\mathbf{K}}^{-1} \boldsymbol{\phi}(\mathbf{x}_ {t'}, t') .
$$

-----------

Another change that has ended up taking much more time than expected is to be more rigorous/structured with the mathematical content, organizing things into theorems/propositions/etc. This has taken much more time than anticipated...So it goes.

---

### Meta-Review · Area_Chair_JMtq · 2023-12-14

**Metareview:**

The authors seeks to identify what properties of score function estimation lead to generalization in diffusion models. Their hypothesis is that the high variance at earlier times of the conditional score in denoising-score matching leads to generalization. They study cases where some of the derivations are analytic. All of the reviewers on this paper were negative for a variety of reasons, such as contradictory evidence in the literature (Stable Target paper), the lack of a real analysis, and generally being incomplete. The authors acknowledge the paper was finished and have made improvements during the rebuttal, but the overall sentiment of the reviewers did not change.

**Justification For Why Not Higher Score:**

The paper feels pretty incomplete and there was no positive support.

**Justification For Why Not Lower Score:**

NA

---

### Decision · Program_Chairs · 2024-01-16

Reject